The use of bat houses as day roosts in macadamia orchards, South Africa

Weier Sina M. 1
Linden Valerie M.G. 1
Grass Ingo ingo.grass@agr.uni-goettingen.de 2
Tscharntke Teja 2
Taylor Peter J. 1 3
1 SARChI Chair on Biodiversity & Change, School of Mathematical & Natural Science, University of Venda , Thohoyandou , Limpopo , South Africa
2 Agroecology, Department of Crop Sciences, University of Goettingen , Göttingen , Germany
3 School of Life Sciences, University of KwaZulu-Natal , Durban , KwaZulu-Natal , South Africa
Livesley Stephen
Electronic publication date: 2019 May 22
Publication date: 2019
Volume: 7
Electronic Location ID: e6954
Received 2018 Oct 12; Accepted 2019 Apr 13
Copyright: ©2019 Weier et al.
Copyright year: 2019
Copyright holder: Weier et al.
License: This is an open access article distributed under the terms of the Creative Commons Attribution License, which permits unrestricted use, distribution, reproduction and adaptation in any medium and for any purpose provided that it is properly attributed. For attribution, the original author(s), title, publication source (PeerJ) and either DOI or URL of the article must be cited.
License URL: https://creativecommons.org/licenses/by/4.0/

Keywords: Artificial roost, Bat boxes, Africa, Chiropteran, Roost site preference

Funding: Europe & South Africa Partnership for Human Development –EUROSA+ the German Academic Exchange Service—DAAD 57371376 57314657 German Federal Government through the Limpopo Living Landscapes project of the SPACES (Science Partnerships for the Assessment of Complex Earth System Processes) the University of Venda and National Research Foundation (NRF) Department of Science and Technology (DST), through the SARChI Research Chair on Biodiversity Value & Change This work was supported by the Subtropical Fruit Association of South Africa (Subtrop) by providing the initial funding for the bat houses, by the Europe & South Africa Partnership for Human Development –EUROSA+; the German Academic Exchange Service- DAAD [grant numbers 57371376 and 57314657]; German Federal Government through the Limpopo Living Landscapes project of the SPACES (Science Partnerships for the Assessment of Complex Earth System Processes); the University of Venda and National Research Foundation (NRF) and Department of Science and Technology (DST), through the SARChI Research Chair on Biodiversity Value & Change in the Vhembe Biosphere Reserve. The funders had no role in study design, data collection and analysis, decision to publish, or preparation of the manuscript.

==============================
The loss of roost sites is one of the major drivers of the worldwide decline in bat populations and roost site preferences, either natural or artificially provided, are not well known for African bat species specifically. In this study we focus on the preference for different artificial roost sites by insectivorous bats in macadamia orchards in northern South Africa. From June 2016 to July 2017 we monitored 31 bat houses, mounted on poles in six macadamia orchards, for presence of bats or other occupants. Twenty-one multi-chambered bat houses of three different designs were erected in sets of three. Additionally, five Rocket boxes, four bat houses in sets of two (painted black and white) and one colony bat house were erected. Bats were counted and visually identified to family or species level. From December 2016 to the end of March 2017 iButtons were installed to record and analyze temperature variation within one set of three bat houses. We related the occupancy of bat houses to the different types of houses and the environmental variables: distance to water, altitude and height of the bat houses above the ground. Overall bat house occupancy was significantly higher in the central bat house, in the set of three, and the black bat house, in the set of two. Mean temperatures differed between houses in the set of three with the central bat house having a significantly higher mean temperature than the houses flanking it. Our study might confirm previous assumptions that the microclimate of bat houses appears to be an important factor influencing occupancy. In conclusion, from the different bat houses tested in this study the designs we assume the warmest and best insulated attracted the most bats. Further research is needed on the preferred microclimate of different bat species, co-habitation within bat houses and the potential importance of altitude and distance to water. Our study provided little variation in both altitude and the distance to water.

Introduction

In Europe, artificial bat roosts have long been tested, particularly in silviculture (Bäumler, 1988; Issel & Issel, 1955; Natuschke, 1960; Schwenke, 1983) which led to a more successful design of artificial roost boxes (Schwenke, 1983). More recently, the value of insectivorous bats for agriculture and the use of bat houses in agricultural landscapes have received growing attention (e.g., Boyles et al., 2011; Puig-Montserrat et al., 2015; Taylor et al., 2018). Nevertheless, an ever growing human population and related land use changes, especially agricultural intensification, have led to a threat of extinction to about one quarter of global bat species (see Figure 1.3 in Voigt & Kingston, 2016; Mickleburgh, Hutson & Racey, 2002; Tilman et al., 2001; Tscharntke et al., 2012). The loss of roost sites caused by land use changes is one of the major drivers of this decline (Mickleburgh, Hutson & Racey, 2002; Park, 2015) and there is a particular lack of knowledge regarding roost site preferences of African bat species (Monadjem et al., 2009; Monadjem et al., 2010a; Taylor, 2000). Given accelerating land use change from natural to agricultural landscapes in the developing world and an assumed decline of South African bat populations (Voigt & Kingston, 2016), proactive management of these populations is indispensable to sustain the ecosystem services they provide (Cumming et al., 2014; Taylor et al., 2017; Tuttle, Kiser & Kiser, 2013). Proactive management of bat populations requires filling knowledge gaps about roost site preferences for African bat species, in particular around intensive agricultural systems (Monadjem et al., 2009; Park, 2015; Taylor, 2000).

Peer-reviewed studies focusing on artificial roost site use by African bat species are non-existent and most studies have been conducted in Europe, followed by North America and Australia (Rueegger, 2016). Summarizing these, bat species seem to have a general preference for large volume and multiple compartment bat houses mounted on poles or houses rather than on trees (Rueegger, 2016). There also seems to be a preference for bat houses built from woodcement although these studies are mostly from Europe (Dodds & Bilston, 2013; Gerell, 1985; Haensel & Tismer, 1999; Rueegger et al., in press). Generally, the microclimate of bat houses (influenced by e.g., insulation, sun exposure and color) seems to be an important factor for bat house occupancy (Fukui et al., 2010; Rueegger, 2016; Shek et al., 2012). Looking at different designs and colors of bat houses, studies suggest that preferences also vary greatly depending on the reproductive state of females (Baranauskas, 2009; Fukui et al., 2010; Flaquer, Torre & Ruiz-Jarillo, 2006; Kerth, Weissmann & König, 2001). Furthermore, many bat species seem to be sensitive to competition for bat houses by other species including birds, social bees, ants and wasps (Baranauskas, 2009; Dodds & Bilston, 2013; Meddings et al., 2011). There is a need for research on artificial roost site use, especially in Africa, and this is the first peer-reviewed study of bat house occupancy in South Africa. Successful bat house design and deployment seem to relate mostly to the climatic region and bat species targeted (Rueegger, 2016). Therefore, the objective of this study was to gain insight into the preference for different artificial roost sites by insectivorous bats in macadamia orchards in South Africa. The main research question was: What are the key features of occupied artificial roost sites? We hypothesize that bat houses providing the warmest microclimate will have higher occupancy.

Material and Methods

Study area and bat house design

The study was conducted in the fruit growing area of Levubu (Limpopo Province, South Africa) which accounts for the second highest production of macadamia nuts in the country (Fig. 1). The climate in the study area is subtropical with approximately 1,000 mm of annual rainfall (Taylor et al., 2017). Dominant land use types other than macadamia are pecan, avocado and banana orchards as well as pine and Eucalyptus spp. plantations (Taylor et al., 2017). The South African Subtropical Growers’ Association (Subtrop) arranged for 21 multi-chambered bat houses to be mounted on poles, in sets of three, in four macadamia orchards in Levubu in 2014. Every set of three houses comprised one 6-Chamber bat house in the middle as well as an Old George bat house and a Nursery bat house to either side (Fig. 2; bat houses supplied by EcoSolutions Pty Ltd, Johannesburg, South Africa). Both the Nursery bat house and the Old George bat house vary slightly in their chamber design. The Old George model is different from the other designs in that several of the partitions between the different chambers are set at an angle (Freed & Falxa, 2010) whereas the Nursery bat house has four chambers, which are progressively shorter in length towards the back of the house (Fig. 2). These bat houses were serviced in May 2016 and five two-chambered Rocket boxes (Fig. 2) were put up close to all but one of the sets of three bat houses (EcoSolutions Pty Ltd). The freestanding Rocket boxes allow the bats to move 360° degrees within the house and therefore choose from a range of temperatures depending on sun exposure. An additional four, four-chambered bat houses, in sets of two (painted black and white respectively in order to provide different microclimates) and one colony bat house were also mounted on poles and erected in March 2016 in two additional orchards (Fig. 2). All bat houses were constructed of wood and mounted on eucalyptus poles (see Table S1 for detailed information). As proposed for temperate climatic regions by the North American bat house research project (Kiser & Kiser, 2004), all bat houses erected for this study have open bottoms. They were also placed near water sources and natural vegetation wherever possible. Except for the single colony bat house, several bat houses were erected at each location as this is proposed to positively influence occupancy by providing different microclimates (Sedgeley, 2001). Alternating between different bat houses might also be necessary for bats to avoid predators and high ectoparasite loads (Lewis, 1995; Reckardt & Kerth, 2006).

Figure 1 Map showing the aerial image of the study area with the location of the different bat houses and the study area in Levubu, Limpopo, South Africa (Google Maps 2018; QGIS version 2.18.11; Map data: AfriGIS (Pty) Ltd., Google)).

(A) shows a detailed example of one study site and (B) the whole study area with the location of each bat house as well as (C) the location of the study area (star) in Levubu, Limpopo, South Africa.

Figure 2 Showing the different bat house designs from the front and below, erected in the study area Levubu, Limpopo, South Africa. Photo credit: SM Weier.

(A) shows the set of three houses (B) shows the set of black and white bat houses, (C) shows the Rocket box design and (D) shows the Colony bat house. The below view showing (E) the Old George, the 6-Chamber and the Nursery bat house (left to right) (F) black and white house (G) Rocket box (H) Colony bat house.

Bat house occupancy

All bat houses were monitored monthly, from June 2016 to July 2017. Once a month we scanned each house for bats or any other occupants such as wasps, birds and bees by reflecting sunlight into the houses using a mirror. This process was kept as short as possible to minimize disturbance to bats. Bats were counted and identified visually to family or species level referring to Monadjem et al. (2010b) and species records of Weier et al. (2018). Bat houses solely occupied by wasps or bees were cleared during maintenance in May 2016. We worked under a permit (No. 001-CPM403-00010) for research on small mammals granted by the Limpopo Department of Economic Development, Environment and Tourism.

From the 1st of December 2016 to the 20th of March 2017, three iButtons (Thermochron, DS1921G-F5) were installed at each of the three entrances of one set of three bat houses to record temperature variation between the bat houses every 1.5 h. While scanning the houses and fitting the iButtons, we tried to avoid touching the bat house or the pole supporting the house to keep disturbance to a minimum (Tuttle, Kiser & Kiser, 2013). We recorded the distance to the closest water source using Google Earth, the cardinal direction of the front of the bat house using a digital compass (Axiomatic Inc., Seoul, South Korea), the altitude above sea level (Global Positioning System waypoints; Bluecover Technologies, Lisbon, Portugal) and the height above ground level of each bat house.

Data analysis

All statistical analyses were conducted with R (version 3.4; R Core Team, 2017). The bat house of the type ‘colony’ was not used for analyses as there was only one bat house of this type. The relatively small sample size of 31 bat houses (with 21 of them erected in sets of three) led to correlation between the type of bat house and the direction they were facing (nonzero entries in the ‘alias’-matrix; (Chambers, Freeny & Heiberger, 1992) so this variable was discontinued in the analysis. We then fitted a model to analyze the relationship of the response variable ‘presence or absence of bats’ and the remaining predictor variables ‘type of bat house’, ‘altitude’, ‘distance to water’ and ‘height of the bat house’ (see Table S2). After testing the model for normal distribution and constant errors variance, we applied a generalized linear mixed model (GLMM) with a binomial distribution (package ‘lme4’; Bates et al. 2015). The variables ‘farm’ and ‘month’ were used as random factors to account for pseudo replication and all the numeric predictor variables were scaled. We used the ‘dredge’ function (package ‘MuMIn’; Barton, 2017) based on the lowest values of the Akaike information criterion (AICc), corrected for a small-sample size, to select the final model. In order to analyze differences in temperature recorded within one of the sets of three houses we fitted a linear mixed-effects model. The model included month as a random effect to account for repeated measures of temperature in time. We subsequently used the Tukey test for multiple comparison (package ‘stats’; (R Core Team, 2017). Additionally, we used the summary statistics base function ‘tapply’ to look at differences in range and mean of temperature (R Core Team, 2017).

Results

From June 2016 to July 2017 we recorded 166 individual bats within the 31 bat houses with a maximum of five individual bats in one bat house (Dataset S1). The highest total numbers of bats were recorded in March and May (average 0.53 bats per house) and the lowest numbers were recorded in August and November (0.3 bats per house). We recorded yellow-bellied house bats (Scotophilus dinganii) on 43 occasions, small plain-faced bats (Vespertilionidae) on nine occasions, free-tailed bats (Molossidae) on 21 occasions and Mauritian tomb bats (Taphozous mauritianus) on three occasions (see Table S3).

Table 1 Final model testing the relationship between the occupancy of bat houses and the different types of bat houses in macadamia orchards, Levubu, South Africa (significance level of bold p < 0.05).

Variable	Estimate	SE	Z	p-Value	AICc	
Null model					388.2	
Type of bathouse		
6-Chamber	1.43	0.41	3.45	0.000		
Black	2.30	0.56	4.08	0.000	364.6	
Nursery box	0.50	0.45	1.10	0.268		
Rocket box	−0.50	0.62	−0.81	0.415		
White	0.57	0.67	0.85	0.394		

Type of bat house

The first five models selected by ‘dredge’, analyzing the relationship of the response variable ‘presence or absence of bats’ with the predictor variable(s) were all within a delta AICc of <2 (see Table S2). After testing all five models, which each retained ‘type of bat house’ as the only significant variable, we decided on the simplest final model to be presence of bats ∼ type of bat house. The probability of bat house occupancy in the macadamia orchards was higher in the central bat house (β = 1.43, SE = 0.41, p < 0.001) in the set of three houses and the black bat house (β = 2.30, SE = 0.56, p < 0.001) in the set of two houses (Table 1 and Fig. 3).

Figure 3 Showing the probability (Confidence intervals of 95%) of a certain type of bat house being occupied with in the study area Levubu, Limpopo, South Africa.

Bat house A (=Old George), bat house B (=6-Chamber) and bat house C (=Nursery) are always set up in sets of three with the 6-Chamber bat house in the middle. The black and white pained bat houses are also erected in sets. Annotated letters show same probability of occupancy (same letter) or a significant difference in occupancy levels (different letter).

Temperature

The linear mixed-effects model showed that there was a significant difference in the mean temperature values between the set of three bat houses, with the central bat house (β = 0.46, SE = 0.16, p = 0.005) being significantly warmer (Dataset S2). The mean temperature was the warmest in the second (central) bat house with 23.52 °C (±5.37 SE) compared to 23.06 °C (±5.18 SE) in the first bat house and 23.13 °C (±4.96 SE) in the third bat house. A post hoc Tukey test showed that there was a significant difference in mean temperature of 0.46 °C between the second, 6-Chamber bat house, and the first (Nursery) bat house (p = 0.005) and a significant difference of 0.39 °C between the third, Old George bat house and the second house (p = 0.020). The temperature range was between a minimum of 12.5 °C and maximum of 40.5 °C in the Nursery bat house and between a minimum of 12 °C and maximum of 41 °C in the second and third bat house (6- Chamber and Old George house). The ambient temperature recorded in Ratombo (about 7 km from the iButtons used in this study) from December to end of March over the last 30 years shows a mean daily minimum of 18–19 °C and a mean daily maximum of 27–28 °C with peaks of up to 35 °C in December (Meteoblue, 2018).

Other bat house occupants

We observed a number of other animals occupying the bat houses both in the presence and absence of bats. We encountered lesser galago (Galago moholi), nests likely to belong to tree squirrel (Paraxerus cepapi) or dormouse (Myoxidae), lizards (Lacertilia), social wasps (Vespidae) and honeybees (Apis). We twice (May 2017 and June 2016) observed yellow-bellied house bats sharing a bat house with an active honeybee hive.

Discussion

The highest (March and May) and lowest (November and August) average numbers of bats recorded during our study overlap the high (December to end of May) and low (June to end of November) seasons of insect pest species occurrence in macadamia orchards (Weier et al., 2018). The study of Weier et al. (2018), conducted in the same study area, shows that bat activity nearly doubled in the high season and increased with Hemipteran abundance. This also supports the suggestion that insectivorous bat species track outbreaks of insect pests (to the macadamia growers) such as certain stinkbug (Hemiptera: Pentatomidae) species (Taylor, Monadjem & Steyn, 2013; Taylor et al., 2017; McCracken et al., 2012). We therefore suggest that colonization of bat houses in and around macadamia orchards was highest in times of high prey availability (Fig. 4).

Figure 4 A Common Slit-faced Bat (Nycteris thebaica) foraging on a green vegetable stinkbug (Nezara viridula) in the study area Levubu, Limpopo, South Africa. Photo credit: ©MerlinTuttle.org.

Yellow-bellied house bats were by far the most frequently recorded species (42 records). This species is naturally tree cavity roosting but well known to utilize anthropogenic structures, particularly attics (Monadjem et al., 2010a). Given the large size of yellow-bellied house bats (weighing up to 37 grams) they might have a competitive advantage over smaller bat species (Monadjem et al., 2010a). However, species displacement in the artificial roosts can only be confirmed by, for example, fitting cameras to the houses as in the study of Kerth, Weissmann & König (2001). Furthermore, it is not yet known what effect the installation of artificial roosts has on the local community composition of bats and if it might lead to displacement of rare species by common species (Russo & Ancillotto, 2015). While we observed a number of other animals in the bat houses during this study, the bat house design did not seem to attract birds such as was reported by Dodds & Bilston (2013). Interestingly, while there are contrasting observations on whether wasps displace bats from bat houses (Rueegger, 2016), we did not observe co-habitation with wasps but with active beehives. Tree squirrels and/or dormice might be able to displace bats as we only found a bat house occupied by bats with a mammal nest present once, in which case the nest appeared to have been abandoned for some time.

The collection of faecal pellets from the artificial roosts (for a parallel project on bat diet) suggests that the occupancy of bat houses is much higher than we recorded during our monthly visits as we often collected faecal pellets under seemingly unoccupied bat houses. While alternating between different roosts is well known, especially for pregnant bats (Kerth, Weissmann & König, 2001; Reckardt & Kerth, 2006) as well as in fission and fusion behaviour (Kerth & König, 1999), our study focused on the use of bat houses as a day roost and different occupancy numbers might be observed when conducting night visits.

As visits were conducted monthly, we could also not exclude the possibility that signs of bat presence such as faeces were washed away by rain or disintegrated in the sun. We therefore did not include pellet counts in our analyses but this should be kept in mind for future studies and might be a possible variable to use for studying winter roost occupancy during the dry months in northern South Africa, especially if data is collected over several years.

The colony bat house remained unoccupied throughout this study. In order to distinguish if this is an effect of the location or the type of bat house, several colony bat houses need to be erected and monitored. We also suggest that the colony bat house might have remained unoccupied because it did not provide any other artificial roost sites in close vicinity so alternating between different bat houses was not possible. The Rocket box design has been particularly successful in the United States and Canada with 62% overall occupancy during a five year survey (Kiser & Kiser, 2004). Therefore, local occupancy of the Rocket boxes might still increase with time (Agnelli et al., 2011). However, one component, which both the Rocket box and the Colony bat house are missing compared to the other designs erected, is a landing board (Fig. 2).

All of the bat houses in this study were freestanding with no direct cover from trees or houses. While we were unable to analyze the influence of the cardinal direction respective to sun exposure on bat house occupancy, we suggest that future studies should consider this variable, particularly if bat houses are mounted onto the walls of houses and receive shadow. It would also be interesting to test bat houses mounted back-to-back to provide additional insulation as proposed by Kiser & Kiser (2004) and to control temperature variation between all houses of different designs.

The distance to water, the altitude above sea level and the height of bat houses did not significantly influence bat house occupancy in our study. While water availability is known to influence bat activity (Crisol-Martínez et al., 2016; Grindal, Morissette & Brigham, 1999; Rainho & Palmeirim, 2011) the bat houses in this study were erected within a range of two to 680 m from the closest water source, all within the known home range of even small plain-faced bats (Monadjem et al., 2010b). We suggest that the distance to water might be more significant in arid regions compared to our high rainfall subtropical study area. There might also be a difference in this response during the dry season that we suggest should be analyzed once a large data set becomes available. The altitude ranged from 607 to 932 m in this study and therefore did not provide for a great variation in temperature between sites.

Our study appears to confirm previous findings that the microclimate of bat houses (e.g., insulation, sun exposure and color) is an important factor influencing bat house occupancy (Fukui et al., 2010; Kerth, Weissmann & König, 2001; Lourenco & Palmeirim, 2004; Sedgeley, 2001; Shek et al., 2012). Bats generally preferred the black houses in the set of black and white as well as the 6-Chamber models which were the central bat houses in the sets of three. The 6-Chamber models were flanked on either side by other houses and had the most chambers of all bat houses, providing additional insulation (Fig. 2). We suggest that this insulation affected the preference by bats rather than the bat house design. It would be worth investigating how occupancy changes if the 6-Chamber model is erected on the exterior of a set of houses. It should also be noted that only one set of three houses was measured with iButtons in our study and temperature results are based on this limited data. Future studies should aim at a higher sample size in order to investigate temperature variances caused by environmental factors and/or positioning of the houses. Because the color black absorbs wavelengths and therefore energy and white reflects them, it is reasonable to assume that the black bat houses in the sets of two are warmer than the white bat houses. Precise information on the temperature differences would provide insight into the preferred microclimate of houses by different bat species.

It should also be noted that we found dead bats in or under bat houses on three occasions. Although we can not make an informed statement as to cause of death, we would advise caution when placing bat houses within orchards which are regularly sprayed with pesticides. We recommend erecting bat houses at the edges of orchards at some distance to the crops which are sprayed.

Conclusions

The bat houses which we assume were the warmest and best insulated and the sets of houses which were mounted freestanding on worked best to attract bats to occupy them. Further research is necessary and should focus on co-habitation and species displacement within the houses. Future studies should also aim to study a greater variation in altitude and distance of bat houses to water. There is vast scope to experiment with different colors, designs and position of bat houses and to focus on preferred microclimate of different bat species, specifically their response to temperature variation within the houses.

Supplemental Information

Dataset S1 Raw data on bathouses including absences

Click here for additional data file.

Dataset S2 Temperature data measured in each of a set of three bathouses

Click here for additional data file.

Dataset S3 Tables for details on bathouse designs and batspecies recorded during the study of bathouses on macadamia orchards, Levubu, Limpopo, South Africa

Click here for additional data file.

We thank Dr. Merlin Tuttle of making the pictures from his visit to northern South Africa available. Many thanks to all macadamia growers in the Levubu study area for their support, in particular Alan Whyte, Herman De Jager, the late Alistair Stewart, Branden Jardim, Jaco Roux, Fritz Ahrens, Piet Muller and Dave Pope. We would also like to thank staff of the Agroecology Department of Crop Sciences of the University of Göttingen as well as Esther Fichtler for their support. We thank Jabu Linden for editing the manuscript. Three reviewers, including Joanna Burger, provided helpful comments on the manuscript.

Additional Information and Declarations

Competing Interests

Author Contributions

Field Study Permissions

Data Availability

The authors declare there are no competing interests.

Sina M. Weier conceived and designed the experiments, performed the experiments, analyzed the data, prepared figures and/or tables, authored or reviewed drafts of the paper, approved the final draft.

Valerie M.G. Linden conceived and designed the experiments, authored or reviewed drafts of the paper, approved the final draft.

Ingo Grass analyzed the data, prepared figures and/or tables, authored or reviewed drafts of the paper, approved the final draft.

Teja Tscharntke authored or reviewed drafts of the paper, approved the final draft.

Peter J. Taylor conceived and designed the experiments, contributed reagents/materials/analysis tools, authored or reviewed drafts of the paper, approved the final draft.

The following information was supplied relating to field study approvals (i.e., approving body and any reference numbers):

The Limpopo Department of Economic Development, Environment and Tourism approved the study (Permit No. 001-CPM403-00010).

The following information was supplied regarding data availability:

The raw data recorded for this study is available in the Supplemental File.

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
