# Peer review of "The use of bat houses as day roosts in macadamia orchards, South Africa"

_PeerJ, doi:10.7717/peerj.6954_

## Round 0.1 · original submission · Major Revisions

Dear authors, I have received three reviews from expert reviewers and am comfortable with recommending major revisions. I encourage you to fully consider the detailed comments, suggestions and expectations of the three reviewers and to provide a thorough and itemised/justified response to all.

To assist you in this I would like to direct you towards what I regards as the key action items:

Please provide a stronger and chronologically true explanation (with a figure) of the bat house locations in each orchard, and where the orchards are in relation to one another (all reviewers commented on this).
Statistical analysis needs to much better detailed and explained, and please consider a table of all possible models be presented with AICc scores.

Please clarify which bat houses were measured with i-button, especially in relation to colour and type, and consider providing information on ambient temperatures and then minimum and maximums as these are likely to drive suitability. Consider and limitations this may present in the interpretation of results in the discussions.

The discussion in its current form is too speculative and should be better grounded in what the data can support. This does not mean that some speculative statements cannot be included, but they must be clearly stated as thus.

Please try to incorporate more contemporary and relevant literature on bat roost ecology.

I look forward to receiving a revised manuscript in the near future.

Finally, bat houses, is two words not one.

Regards
Steve

Reviewer 1 ·

Basic reporting

This is an important study testing the factors that influence bat houses occupancy in macadamia orchards. No comments regarding the English level used. Some of the literature used was quite repetitive and some times (Rügger et al. 2016), the reference list was longer than the sentence that was referring to (line 58-59).
Some of the figures and tables need to be appropriately described such as which are the dimensions (mm or cm) of the artificial roost.
Regarding to the raw data shared, I wasn’t able to opened, it was illegible.

Experimental design

Even thought that they have not used a large number bat boxes, they obtained good results in terms of what type of bat boxes do they prefer most.
Seems quite confusing for me when and how the bat boxes were erected, I will suggest a chronological explanation that may also be accompanied with a figure showing the location of each type in each orchard.
Another methodological concern is the small sample size monitored with the Ibuttons not knowing the temperature of the rest of the bat boxes. Addressing this potential criticism in the discussion would be advisable.

Validity of the findings

Statistical analysis should be improved explaining the packages they used in R and specifying the summary statistics used.
Also adding the ANOVA obtained to the supplementary material will be helpful.
Seems quite confusing the way to show the results regarding the statistical models used, better to show them all in supplementary materials and just comment the selected one
However, the conclusions are well stated and linked to the hypothesis.

Annotated reviews are not available for download in order to protect the identity of reviewers who chose to remain anonymous.

Reviewer 2 ·

Basic reporting

No comments; see general comments.

Experimental design

No comments; see general comments.

Validity of the findings

No comments; see general comments.

Additional comments

The study entitle “The use of bathouses as day roosts in macadamia orchards, South Africa” investigates factors associated with the structure and environment of bat houses influence bat occupancy in a region where little is known about important roost characteristics. I believe this study is an excellent contribution to the field of using artificial structures for conservation in agricultural areas, especially for species that provide vital ecosystem services through crop pest control. Overall, I found the manuscript to be well written and the study methods and analyses to be robust. I only have minor comments on the content in the manuscript as outlined below:
1. Figure 1: Could you add a zoomed in aerial image of the study location showing: 1) where the orchards were located within regard to one another, and 2) where the bat houses where located within those orchard sites? This would allow the reader to better understand the distribution of the bat houses in each of the orchards (since different numbers of the types of bat houses are placed in each orchard site) and get a better sense of proximity to water and other landscape features.
2. Lines 135-137: I’m a little confused which three bat houses in each of the orchard sites received an IButton unit. Is it the three multiple chamber houses but not the black/white ones, the rocket ones or the colony one? Please clarify. (Note: once I read the results it did become clear that it was the 3-set of bat houses, but perhaps you could clarify in the methods still)
3. I agree with the idea that type of bat house may be confounded with temperature and insulation of the central bat house, since that was the warmest bat house (most likely due to its location in the set up and had more chambers). What this made me realize though was I didn’t see (or may have missed) what the type of bat house was for the black/white pairing. Can you add this information somewhere or make it more apparent, if I missed it? If the black/white bat houses were the six chamber house, then perhaps temperature is the most important, but of course factors affecting temperature are also the number of chambers which help insulate it.
4. Why do you report 166 individuals in the results and 220 individuals in the discussion? Please address this discrepancy.
5. Minor: should it be bathouse or bat house? It’s called the North American Bat House Program, so perhaps separating the words is most appropriate?

·

Basic reporting

The manuscript was generally well written but it could be more concise and checked for the occasional incorrect use of tense. While a minor matter of personal preference, and not necessary to change, my preference is to reference literature without the extra words (e.g., “According to…”, “The review of…”) as this is implied. Instead I acknowledge the published literature with appropriate citations but very rarely noting the authors in anything other than parentheses. For instance Lines 66-68 could be reduced to “Peer-reviewed studies focusing on artificial roost site use by African bat species are literally non-existent (Rugger et al. 2016).” In fact, the three sentences that speak to the review of Rugger et al. could be combined into 1-2 sentences.

Lines 60-65 - the link in the first paragraph of global bat decline to bats being useful for pest control to the lack of knowledge of bat roost preferences and artificial roosts is not clear. While implicitly understood, perhaps this could be made more explicit?

While it is understandable that there is limited published literature on bat house use by bats, I think the authors rely too much on dated grey literature when there might be published alternatives, even if dated. For example, there is reference to Kiser & Kiser (2004) to substantiate comments on roost fidelity and parasite loading when instead “Lewis, S.E., 1995. Roost Fidelity of Bats: A Review. J. Mammal. 76, 481–496.” could be used. If this study was indeed meant to be an extension, build upon, or be a confirmation of the Bat Conservation International newsletter findings, perhaps it could have been framed that way?

Experimental design

While the overall concept of the research is important and could help inform bat conservation - determining the key features of artificial roosts used by bats - I am not convinced that this study/analysis was sufficient to disentangle key features from potentially confounding factors.

While not incorrect to use the term “occupancy”, this term is fairly unambiguously used to describe occupancy modeling (e.g., MacKenzie et al. 2005). Perhaps “bat use” instead?
The authors acknowledge that bats have occupied the boxes on days when no bats were physically present. Was the fact that bats had been present between monthly visits noted and included in the analysis? If bat use was recorded monthly, wouldn’t the present of faeces be an indication of presence during the month and be worthy of inclusion into the analysis? Perhaps as an overall measure of “bat” use - i.e., aggregate all sightings and indications of use as “use” for the month and re-run the analysis, particularly as there aren’t any fine-scale temporal covariates?

I think the statistical methods could be written more clearly. Could the full model be written out for clarification? I’m assuming the full model was something like: Bat presence ~ Bat House Type + Altitude + Distance to Water + Height of Bat House + (1|Farm) + (1|Month). Is this correct? I wonder if “Time Since Installed” should be a fixed factor as literature supports (and this manuscript acknowledges) that bat use of artificial roosts generally increases over time. The fact that 21 bat houses (in sets of 3) were installed years earlier than the other bat houses and the remaining houses were all installed in May or March 2016 makes me wonder if the reason that the colony house had no bats was a confound of only having 1 replicate of that type and it being installed in March 2016.

I’m not quite sure why the data for Bat House Type “Colony” was removed - if the model is set up as I wrote it then there is no reason for data to be removed. I’m also not clear on why Farm and Month are used as random factors. The authors acknowledge that there might be some sort of temporal trend but then use Month as a random factor. Why not convert month into a continuous variable (such as a sine/cosine function) and keep it in the model as a fixed factor to test if there is a temporal trend? I’m also a little concerned with the use of the “dredge” MuMIn function and the low sample sizes. Considering all of the tests that are run using dredge, might the few statistically significant results be spurious (i.e., if you run 100 tests you should expect 5 of them to show significance from random chance)?

Table 1 is not very useful. It’s not clear from the methods/results how the models were run (see comments above) and thus it’s not clear if this is the best top model (as suggested by the AICc score, which is the same for all variables and thus redundant in this table) or if these were competing models, which is normally what would be presented with AICc scores. Could a table of all possible models be presented with AICc scores or delta AIC to give the reader an idea of how many models were run and how “good” the top model is in relation to all other models?
Did the authors run a null model and full model to compare against their top model? It would be useful to see the results, especially if the top model is significantly better than the null/full (i.e., comparing log-likelihoods using something like the anova() function in R). In addition, have the authors thought about calculating the evidence ratio to determine the strength of evidence for each variable? It might be valuable for the authors to review Burnham et al 2011 prior to a revision of the analyses. (http://www.ericlwalters.org/Burnham_etal_2011.pdf).

I’m not clear on the purpose of the iButtons recording temperature in only 3 (adjacent?) bat houses. What was the ambient temperature? Is mean temperature something that is truly important to bats? I would imagine it might be the maximum/minimum/difference from ambient that might be the most useful as houses that are too hot/cold might be detrimental to bat health. Can the authors provide a rationale for why they chose mean temperature and justification how this small replicate test can provide insight into bat house preferences? Also, I would revise line 190 as p=0.55 is not a “marginal significant difference”.

Is it possible to include a figure of the study area/sampling design? Perhaps the current study area figure (Fig 1) could be included as a small inset to orient the reader to the region with the main figure showing the placement of box types in relation to one another and to the other farms. This would help the reader understand how nested/confounded the experiment is. It would also be quite useful to have some sort of symbol for each of the different type of box and have the size of the symbol proportional to the number of bats detected in the box. I found it very confusing to try and remember the experimental set-up and number of occurrences in each bat box type from reading the manuscript in its current form.

What is the landscape surrounding the farms like? From Figure 2 it seems that it might be treed, although this could just be my ignorance of what macadamia orchards look like. If it is treed do you think this might influence bat use of artificial roosts? I.e., will you be less likely to have bats roosting in the bat houses if surrounded by natural roosts and, if yes, is the surrounding habitat consistent around all farms? This relates back to my comment above suggesting a figure to help the reader understand the layout of the sampling design and if there might be confounding factors.

While a nice picture, Fig 4 seems irrelevant to the study as there was no research relating bat house use to insect availability or insect observations/data of any kind.

Validity of the findings

Unfortunately, I did not find that the data was robust, statistically sound or adequately controlled. Nor did I think that the conclusions were limited to supporting results.

Much of the “suggested” in the discussion might be better framed as author speculation. While interesting thoughts, I might argue that the research does not actually suggest these findings but the authors are speculating based on the findings and/or observations as much of the discussion relates to the observed data rather than results from any statistical tests. For example, is the conclusion related to bat preferences of warmer houses accurate (lines 207-208)? From my reading of the methods I thought that temperature was only recorded in one set of 3 boxes, but not in the set of white/black boxes (lines 134-137). Please clarify the methods. Assuming there was only temperature recorded in the set of three bat boxes, how can there be the conclusion that bats preferred warmer temperatures? This is not substantiated by the findings of bats occurring more frequently in the central bat house and in the black bat houses as temperature was not measured in all the boxes nor related to bat house use in any statistical test.

Additional comments

Other comments: Is bathouse a word? I would write it out as two words or bat box, as is typically written.

Line 164 notes that there were 166 individual bats observed while line 203 notes 220 individual bats. Were 166 or 220 bats observed?

---

## Round 0.2 · Major Revisions

Dear Sina and co-authors, thank you for continuing to improve this manuscript. I greatly appreciate your persistence and the input of the reviewers.

There is still a need to improve the writing style, paragraph structure and flow throughout the manuscript - this may require you to seek the input of an additional friendly colleague who can provide a fresh set of experienced eyes and editing. It is not the responsibility of the reviewers or journal staff to do this for you. Both reviewers remarked on this critical clarity of communication issue. However, reviewer 2 has provided some detailed guidance through the introduction, but this needs to be applied throughout the remainder of the manuscript.

On other specific and important issues, a table is requested that provides the model structures and AICc for the top ranked models as this is still ambiguous. Also, attempt in the discussion to clearly state if the findings of your study are consistent with previous studies: if not, explain why your study may be producing different findings; if so, discuss how your study mirrors or builds upon the consensus of understanding. This will better place your study in the wider and international bat roost ecology literature.

Please also pay close attention to respond to the other detailed comments of the two reviewers.

I look forward to receiving the final revised version of your manuscript.

Regards
Steve

Reviewer 2 ·

Basic reporting

no comment

Experimental design

no comment

Validity of the findings

no comment

Additional comments

In general, the revision to the manuscript helped address many of my previous comments on the manuscript. The new figure 1 greatly helped me understand the set-up of the study. Overall, I think this study provides insights into management/conservation methods using artificial roost structures, and is an important contribution to that field. My only issues with the current version of the manuscript is with the writing style and some-what sloppy formatting. Below I’ve listed examples and places for improvement.
1) Abstract: Lines 51-53 read a bit awkwardly to me. Its important that this sentence is edited as this is a main take-home result. Do you mean “of the different bats houses tested in the study, the houses that had characteristics that made them the warmest and best insulated—which were erected in sets—attracted the most bats”? Do you need to mention that they are freestanding on poles? Because they were all freestanding.
2) In a number of areas, there are one-sentenced paragraphs (Lines 75-77, 99-100, 189-192, 238-241). In most of these cases, I do not see justification to keep them as a single sentence. I recommend incorporating those single-sentence paragraphs into another paragraph to make your points clear. Also, I wasn’t sure if the authors meant to have these sentences be a stand-alone paragraph as the formatting was not consistent throughout the article, with a space inconsistently occurring between paragraphs.
3) Lines 84-85: I suggest changing “Generally, the microclimate of bat houses, respectively insulation, sun exposure and color seems to be an important factor influencing bat house occupancy” to “Generally, the microclimate of bat houses (e.g., insulation, sun exposure and color) seems to be an important factor influencing bat house occupancy.” Because respectively is miss-used in this sentence and the original sentence is complex and difficult to read.
4) Figure 1 is very helpful in understanding the study set up. But, I would like to see the quality of that figure improved in terms of how its formatted, especially with regard to the aerial image used in part (a) and the map used in part (c). In the map in part (c), it is difficult to notice the tiny circle (almost looks like a lake on the map) and seems odd that its not inset within the part (a) image.
5) In a number of places, “respectively” should be changed to “respective to” (e.g., line 122, 250, 313).
6) There are still a couple of places with “bathouse”, not “bat house”. I always read it as “bath house” (e.g., Line 155, 188).
7) The first paragraph in general repeats what is stated in the results and then what is later stated in the discussion. I would remove it since it doesn’t get to the interpretation of the results until paragraphs 2 and 3.
8) Line 276: affect, not effect

·

Basic reporting

The manuscript could still use some English editing/grammar assistance to read more clearly - I make a few suggestions below (I stopped after the introduction unless the meaning of the sentence was ambiguous) but the entire manuscript would benefit from a thorough editorial review for a more concise manuscript, clearly emphasising the findings and implications of those findings. For example, the main results/key implications are not immediately apparent from the abstract; instead the reader has to re-read sentences and makes their own inferences to determine how the results from this study relate to roost site preferences of African bats generally. In addition, the manuscript could be structured in such a way to emphasise the main points - I recommend following the suggestions in this blog (https://dynamicecology.wordpress.com/2016/02/24/the-5-pivotal-paragraphs-in-a-paper/). Lastly, the decision of when to create a new paragraph seems strange to me. I strive to write so that each paragraph conveys one key message, but with all of the necessary information for that message. Perhaps it’s a formatting issue, but it seems odd to me that some sentences are pulled out as an entire paragraph when they fit with the preceding paragraph(s) - e.g., lines 74-76 comprise one sentence and one paragraph that seems to be floating between two paragraphs when all three could be reduced to one. Lastly, the reference formatting and use of spaces, or lack thereof, with “&” between references with two authors, should be checked. In some cases the reference list is unnecessarily long - e.g., lines 58-60 list 9 references when the most relevant 2-3 could be listed with the use of “e.g.”.
Line 24-26 - awkward sentence, should be teased into two sentences as I’m not sure it’s grammatically correct. Do you mean something like: “The loss of roost sites is one of the major drivers of this decline; roost site preferences, either natural or artificial, are not well known for bat species globally, or African species specifically.”
Line 28 - incorrect use of a preposition, should be “mounted on poles in six macadamia orchards”
Line 36-38 - could be more concise, perhaps “…to the environmental variables: distance to water, altitude, height, and difference in mean temperature…”
Line 37-39 - “Bat house occupancy was significantly higher in the central bat house, in the set of three, and the black bat house, in the set of two, compared to all other bat houses” - is this what you mean? Or do you mean that compared only to their neighbouring houses these two houses had higher bat occupancy? It’s not currently clear.
Line 39-41 - “Mean temperatures differed between houses in the set of three; the mean temperature of the central bat house was 0.46°C (± ? SE) higher than the first bat house, and no different than the third bat house.” - is this what you mean? It’s not immediately clear the importance of this finding - is it that mean temperatures were higher/lower in the bat house more/less exposed to sun and thus the implications for others installing bat houses is???? Also, missing some measure of precision, such as standard error, around the mean.
Line 42 - missing species “most recorded species and …”
Line 44 - “appear to positively influence bat house occupancy” can reduce for conciseness
Line 50 - missing comma “species, especially”
Line 52 - why is future research needed on the potential importance of altitude and distance to water? It’s not clear from your abstract - perhaps it’s worth noting that the only environmental variable that influenced bat house occupancy in your study was mean temperature, but that this may be an artefact of little variation in the other variables, and thus future studies are needed to directly test these potentially important factors - or something similar.
Line 70-73 - poor sentence structure; perhaps “Proactive management of bat populations requires filling existing knowledge gaps on roost site preferences for African bat species….” Etc
Line 80 - “built”, also this sentence is unnecessarily long
Line 84 - grammatically incorrect - “seem to be important factors”
Line 97 - what is meant by “warm” - can the hypothesis be more specific?
Line 98 - consider changing “do particularly well” to something more explicit, such as “will have higher bat occupancy”, otherwise the hypothesis is unclear and subjective.
Line 119 - “respectively” is out of place, revise the sentence to concisely state what is meant.
Line 121 - “were also mounted”, “one colony bat house”
Line 131 - poorly constructed sentence - including a misplaced modifier, “unless weather prevented access” reads like the bees were unable to access the house in poor weather
Line 136/137 - perhaps “to minimise disturbance”
Line 143 - delete the extra space between “one set”
Line 145 - do you mean that temperature was recorded hourly or every 1.5 hours?
Line 148 - cardinal direction facing what?
Line 149 - is the height on poles the same as the height above the ground? Perhaps revise to clearly convey the importance of the measurement, as I imagine it is the height above the ground that was the interesting factor, not necessarily how high the pole was (i.e., what if the pole was mounted on something - is it the height above ground of on the pole that matters?).
Line 153 - “bat house”
Line 154 - “led”; is the point that all covariates were tested for collinearity and correlated variables (including VIF) above a certain level were removed. If so, what was the correlation threshold?
Line 156 - “were facing”
Line 217-218 - mean monthly bat counts? It’s not clear what is meant by average number of bats, was it the mean number of bats recorded in each bat house? Considering the start of the sentence is the total number of bats observed I am thinking it should be the highest total number of bats was recorded in March and May, and the lowest recorded in August and November.
Line 315 - “advise” rather than advice

Experimental design

Line 160 - fit a single model to analyze the relationship for each type of bat house? I appreciate the revision to better explain the model but still don’t think it’s completely clear on if it’s one single model or multiple models for each type of bat house. (Also it’s “fit” not “fitted” the model). If one model, then line 166 should read “final model”.
Lines 182-186 - the statistical methods are not clear and thus the results are not clear. I had concerns with the methods in my last review and while the comments provided were helpful, there is still a lack of clarity in the manuscript text. I am assuming that with the dredge function all possible combinations models were run using different combinations of the covariates. These models were ranked by AICc and the model with the lowest AICc was chosen as the top model, or multiple models within 2 AICc were chosen as per Burnham and Anderson (2002). It would be helpful if a table (perhaps the one referred to on line 184) was provided listing the top 5-10 models, the null model and the full model, all with their AICc values to provide the reader of an understanding of what covariates were included in the top models and their relative ranking. It’s unclear if the authors arbitrarily chose the final model because “type of bat house” was in each of the 5 “top” models within 2 AICc of each other or because it was the model with the lowest AICc. Also, it reads as though the authors chose the covariate with a p-value < 0.05, rather than ranking by AICc (without looking at p-values) to determine the covariate(s) in the “top” model.
Line 184 - Supplementary Table 2 was not made available, did the authors mean to refer to Table 1? If so, all five models and their respective AICc values were not listed but rather one AICc was listed for all variables, which I take to assume one model. Unclear regardless.
Line 187-189 - Because the previous paragraph was not clearly written I am not sure about these results. If multiple bat houses were part of a categorical variable, then what type of bat house is the reference category? If univariate models were run, please state this explicitly.
Lines 196-205 - Please clarify how hourly (or 1.5 hour) temperature data is independent and not auto-correlated. I am concerned that the significant p-values are an artefact of the high number of (correlated) samples and may (or may not, difficult to know) be spurious. If averaging min/max/mean values over a 24 hour period or each night (when bats are roosting) the worry of independence may be lessened. In any case, some measure of precision should be reported alongside the means (standard error or standard deviation). Perhaps using a time-series type of model and/or GAM to analyse the temperature data?
Table 1 - what is the reference category for bat house? Also, the caption has “bathouses” rather than “bat houses”

Validity of the findings

Line 220-221 - I don’t disagree that bats may be more likely to prefer a warm bat house but I am still not convinced that this conclusion/assumption comes from the findings. Could the authors more clearly make the connection? Perhaps it would be clearer if the results were presented more clearly.
The above comment could be made multiple times over in the discussion. The study either does or does not confirm previous assumptions (line 235) - I suggest explicitly stating if the study findings are consistent with previous studies, and if not, explain why there might be a difference or if yes, discuss how this study corroborates findings from other studies.
Line 283-284 - was altitude being considered as a proxy for temperature? This suggestion does not seem substantiated by findings in this study as this link has not been mentioned before, i.e., when altitude was introduced as a potentially important factor influencing bat house occupancy.

Additional comments

The objective of the study was to determine artificial roost preferences for insectivorous bats in macadamia orchards in South Africa. This is an important study, especially considering the global decline in bat populations and the need to find proactive management solutions to maintain and conserve bat populations. This is my second review of this manuscript - the authors have thoughtfully responded to the comments provided by all reviewers. However, I feel that the manuscript could use further revision (better explanation of the analysis and general re-writing to concisely convey the salient points) to more clearly and explicitly present the findings and provide implications beyond bat preferences for artificial roosts in macadamia orchards in South Africa.

---

## Round 0.3 · accepted · Accept

Congratulations on a comprehensive and considerate revision of your manuscript in response to the two reviewers detailed comments and suggestions. I am very happy to recommend this manuscript for acceptance in PeerJ.

There is one oversight in the revision which is the missing double ee in Ruegger N (in press). Please correct.

Otherwise, well done and thank you for your patience and persistence through the review process.

Regards
Steve

#